# Chasing Better Deep Image Priors between Over- and Under-parameterization

**Qiming Wu**                                                                    *qimingwu@cs.ucsb.edu*
*University of California, Santa Barbara*

**Xiaohan Chen**                                                          *xiaohan.chen@alibaba-inc.com*
*Decision Intelligence Lab, Damo Academy, Alibaba Group (U.S.)*

**Yifan Jiang**                                                                *yifanjiang97@utexas.edu*
*University of Texas at Austin*

**Zhangyang Wang**                                                              *atlaswang@utexas.edu*
*University of Texas at Austin*

**Reviewed on OpenReview:** *https://openreview.net/forum?id=EwJJks2cSa*

## Abstract

Deep Neural Networks (DNNs) are well-known to act as **over-parameterized** deep image priors (DIP) that regularize various image inverse problems. Meanwhile, researchers also proposed extremely compact, **under-parameterized** image priors (e.g., deep decoder) that are strikingly competent for image restoration too, despite a loss of accuracy. These two extremes push us to think whether there exists a better solution in the middle: *between over- and under-parameterized image priors, can one identify "intermediate" parameterized image priors that achieve better trade-offs between performance, efficiency, and even preserving strong transferability?* Drawing inspirations from the lottery ticket hypothesis (LTH), we conjecture and study a novel "lottery image prior" (**LIP**) by exploiting DNN inherent sparsity, stated as: *given an over-parameterized DNN-based image prior, it will contain a sparse subnetwork that can be trained in isolation, to match the original DNN's performance when being applied as a prior to various image inverse problems.* Our results validate the superiority of LIPs: we can successfully locate the LIP subnetworks from over-parameterized DIPs at substantial sparsity ranges. Those LIP subnetworks significantly outperform deep decoders under comparably compact model sizes (by often fully preserving the effectiveness of their over-parameterized counterparts), and they also possess high transferability across different images as well as restoration task types. Besides, we also extend LIP to compressive sensing image reconstruction, where a *pre-trained* GAN generator is used as the prior (in contrast to *untrained* DIP or deep decoder), and confirm its validity in this setting too. To our best knowledge, this is the first time that LTH is demonstrated to be relevant in the context of inverse problems or image priors. Codes are available at https://github.com/VITA-Group/Chasing-Better-DIPs.

## 1 Introduction

Deep neural networks (DNNs) have been powerful tools for solving various image inverse problems such as denoising Zhang et al. (2017); Guo et al. (2019); Lehtinen et al. (2018); Jiang et al. (2022), inpainting Pathak et al. (2016); Yu et al. (2018; 2019b), and super-resolution Ledig et al. (2017); Lim et al. (2017); Zhang et al. (2018). Conventional wisdom believes that is owing to DNNs' universal approximation ability and learning from massive training data. However, a recent study Ulyanov et al. (2018) discovered that degraded images can be restored independently using randomly initialized and untrained convolutional neural networks

(CNNs) without the supervised training phase, which is called *Deep Image Prior* (DIP). A series of works Cheng et al. (2019); Gandelsman et al. (2019) have been proposed to improve CNN-based DIPs, which indicates that specific architectures of CNNs have the inductive bias to represent and generate natural images well.

Despite the advantageous performance, most DIP methods use highly **over-parameterized** CNNs with a massive number of parameters (we show this in Table 3 in appendices). Over-parameterization causes computational inefficiency and overfitting (and thus proneness) to noises. This trend has naturally invited the curious question: *does DIP have to be heavily parameterized?* That question is partially answered by Heckel & Hand (2018) by proposing the first **under-parameterized**, non-convolutional neural network for DIP named "deep decoder". Deep decoder shares across all pixels a linear combination over feature channels in each layer and thus has an extremely compact parameterization. Thanks to its under-parameterization, deep decoder alleviates the "noise overfitting noise" in DIP. However, the empirical performance of deep decoder is often not on par with overparameterized DIP models, especially on tasks like inpainting and super-resolution. This is potentially attributed to the extremely restricted capacity of deep decoder from the very beginning. The dissatisfaction on both extremes, that is using either over-parameterized CNN DIPs or under-parameterized deep decoders, pushes us to think if there is a better "middle ground": *Between over- and under-parameterized image priors, can one identify "intermediate" parameterization better trade-offs between performance, efficiency, as well as even preserving strong transferability?*

In this work, we adopt *sparsity* as our main tool for the above question. We seek more compactly parameterized subnetworks by pruning superfluous parameters from the dense over-parameterized CNN DIPs (overview of our work paradigm is shown in Figure 1), essentially viewing pruning as a way to smoothly and flexibly "interpolate" between over- and under-parameterization. We have several motivations for choosing sparsity and pruning from over-parameterization. On one hand, compared with dense and overparameterized DIP models, sparsity saves computations during inference and DIP fitting. Moreover, sparsity can serve as an effective prior that regularizes DNNs to have better robustness to noise Chen et al. (2022), that is important for DIP which is tasked to distinguish image content from noise. These two blessings combined make sparsity a promising "win-win" for not only DIP efficiency but also performance. On the other hand, unlike deep decoder which sticks to a compact design, exploiting sparsity follows a different "first redundant then compact" training route, which is widely found to enhance performance compared to training a compact model directly from scratch Zhou et al. (2020). Since overparameterized (especially wide) DNNs have smoother loss surfaces while smaller ones have more rugged landscapes, starting from overparameterization can ease the training difficulty Safran et al. (2021) and may particularly help the "chaotic" early stage of training Frankle et al. (2020c).

The recently emerged *Lottery Ticket Hypothesis* (LTH) Frankle & Carbin (2018); Frankle et al. (2020a) suggests that every dense DNN has an extremely sparse "matching subnetwork", that can be trained in isolation to match the original dense DNN's accuracy. While the vanilla LTH studies training from random scratch, the latest works also extend similar findings to fine-tune the pre-trained models Chen et al. (2020a; 2021b). LTH has widespread success in image classification, language modeling, reinforcement learning and multi-modal learning, e.g., Yu et al. (2019a); Renda et al. (2020); Chen et al. (2020a); Gan et al. (2021). Drawing inspirations from the LTH literature, we conjecture and empirically study a novel "lottery image prior" (**LIP**), stated as:

> *Given an (untrained or trained) over-parameterized DIP, it will have a sparse subnetwork that can be trained in isolation, to match the original DIP's performance when being applied as a prior to regularizing various image inverse problems. Moreover, its performance shall surpass under-parameterized priors of similar parameter counts.*

Diving into this question has two-fold appeals. On the algorithmic side, as the first attempt to investigate LTH in the DIP scenario, it could help us understand how the topology and connectivity of CNN architectures encode natural image priors, and whether "overparameterization + sparsity" make the best DIP recipe. On the practical side, the affirmative answer to this question can lead to finding compact DIP models that are more performant than the deep decoder, hence yielding more computationally efficient solutions for image inverse problems without sacrificing performance.

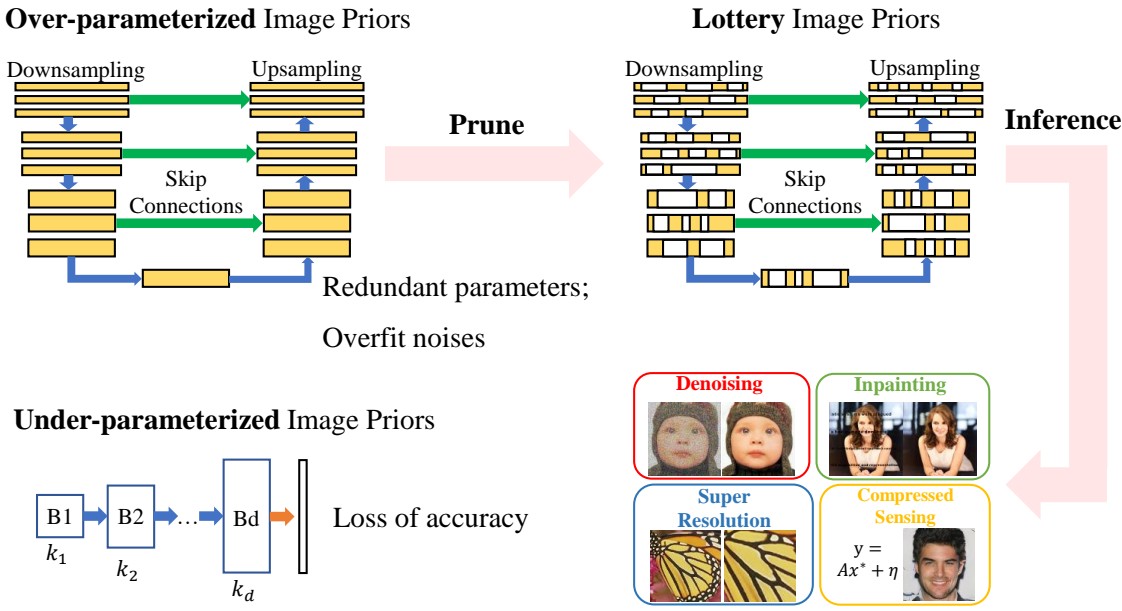

Figure 1: Overview of our work. Between the **over-** and **under-**parameterized image priors, we aim to find the sparse matching networks with better performances, efficiency and strong transferability.

## 1.1 Summary of Contributions

We now state our intended contributions from two angles: how our work might affect the deep image prior research (application), and the lottery ticket hypothesis research (methodology). To avoid misunderstandings, we want to state again: efficiency is definitely not the key target or motivation. Our key target instead is to verify the existence of a "compact" DIP model that is free from severe overfitting but at the same time different from the deep decoder. We want it partially because the deep decoder is bad in terms of performance (e.g., inpainting and super-resolution). Efficiency is also a natural byproduct that results from pruning and sparsity. Improving efficiency could reduce the single-image inference time of DIP to accelerate restoration and reconstruction, which could be helpful in latency-sensitive applications.

For DIP research community, In between over- and under-parameterized image prior models, we aim to use sparsity to find an intermediate parameterized network. Specifically, compared to **over-parameterized** CNN DIPs:

- We successfully locate the "matching subnetworks" [1] from over-parameterized DIP models by iterative magnitude pruning. Their prevailing existence demonstrates that sparsity, as arguably the most classical natural image prior, remains relevant as a component in DIPs.

- Beyond "matching", our found LIP subnetworks can even outperform DIP models on several image restoration tasks (shown in Figure 2), which is in line with the previous findings that the sparsity can enhance DNN robustness to noise and degradations Chen et al. (2022).

- Also, the sparse LIP subnetworks are found with powerful transferability across both test images and restoration tasks. Importantly, that amortizes the extra cost of finding the subnetwork per DIP network, by extensively reusing the found sparse mask. It is also neatly aligned with the recent study of LTH transferability Redman et al. (2022).

Meanwhile, regarding **under-parameterized** image priors such as the deep decoder:

---

[1]A matching subnetwork Frankle & Carbin (2018) is defined as a pruned subnetwork whose performance can match the original dense one, when separately trained from scratch

- Our found LIP subnetworks perform remarkably better than the deep decoder of comparable parameter counts, regardless of test images or restoration tasks. It strongly signifies that pruning from over-parameterization is more promising than sticking to under-parameterization: perhaps the first time indicated in the DIP field up to our best knowledge.

- Besides untrained DIP scenarios (over- and under-parameterized), we extend our method to the pre-trained GAN generator as an image prior used in compressive sensing, which further validates the prevalence of LIPs. It is beyond the existing scope of deep decoder.

For the LTH research community, studying this new LIP problem is also **NOT** a naive extension from the existing LTH methods, owing to several technical barriers that we have to cross: We summarize the contributions as follows:

- Till now, LTH has not been demonstrated for image inverse problems or DNN-based priors, to our best knowledge. Most LTH works studied discriminative tasks, with few exceptions Chen et al. (2021d). It is therefore uncertain whether high-sparsity DNN is still viable for low-level vision tasks such as DIPs. Interestingly though, despite that we demonstrate the existence of LIP, we also find that the LIP matching subnetworks cannot directly transfer to high-level vision tasks such as classification.

- Existing LTH works typically require a training set to locate the sparse subnetwork. In stark contrast, owing to the nature of DIP, we train a DNN to overfit one specific image. While a handful of papers studied LTH in "data diet" settings Chen et al. (2021a); Paul et al. (2022), this paper pushes the extreme to "one-shot lottery ticket finding" for the first time. We also develop a multi-image variant of LIP which boosts performance in the cross-domain fitting.

- DIP fitting practically relies on early-stopping to avoid over-fitting noise in images. The same risk exists in our iterative pruning, and we demonstrate how to robustly find the sparse matching subnetwork from one noisy image, without needing a clean reference image.

## 2 Background Work

### 2.1 Over-parameterized Image Priors

Despite CNNs' tremendous success on various imaging tasks, their outstanding performance is often attributed to massive data-driven learning. DIP Ulyanov et al. (2018) pioneered to show that CNN architecture alone has captured important natural image priors: by over-fitting a randomly initialized untrained CNN to a single degraded image (plus some early stopping), it can restore the clean output without accessing ground truth. Follow-up work Mataev et al. (2019) strengths DIP performance by incorporating it with the regularization by denoising (RED) framework and a series of works Mastan & Raman (2020; 2021) use the contextual feature learning method to achieve the same goal of DIP. Besides natural image restoration, DIP was successfully applied to PET image reconstruction Gong et al. (2018), dynamic magnetic resonance imaging Jin et al. (2019), unsupervised image decomposition Gandelsman et al. (2019) and quantitative phase imaging Yang et al. (2021).

There have been several efforts toward customizing DIP network architectures. Liu et al. (2019) extends the DIP framework with total variation (TV) and this combination leads to considerable performance gains. Jo et al. (2021) further propose the "stochastic temporal ensemble (STE)" method to prevent DIP models from overfitting noises and thus improving performances. Chen et al. (2020c) proposed a neural architecture search (NAS) algorithm, which searches for an optimal DIP neural architecture from a search space of upsampling cell and residual connections. The searched 'NAS-DIP' model can be reused across images and tasks. Arican et al. (2022) observe that different images and restoration tasks often prefer different architectures, and hence design the image-specific NAS to find an optimal DIP network architecture for each specific image. While our work also pursues better DIPs, it substantially differs from those prior arts in terms of model compactness (approaching the under-parameterization end) and reusability. The simple pruning-based recipe also overcomes any extra design hassle (search space or algorithm) caused by NAS. We compare the results with them in the paragraph "**Comparisons with NAS-DIP and ISNAS-DIP models**" in Appendix B.

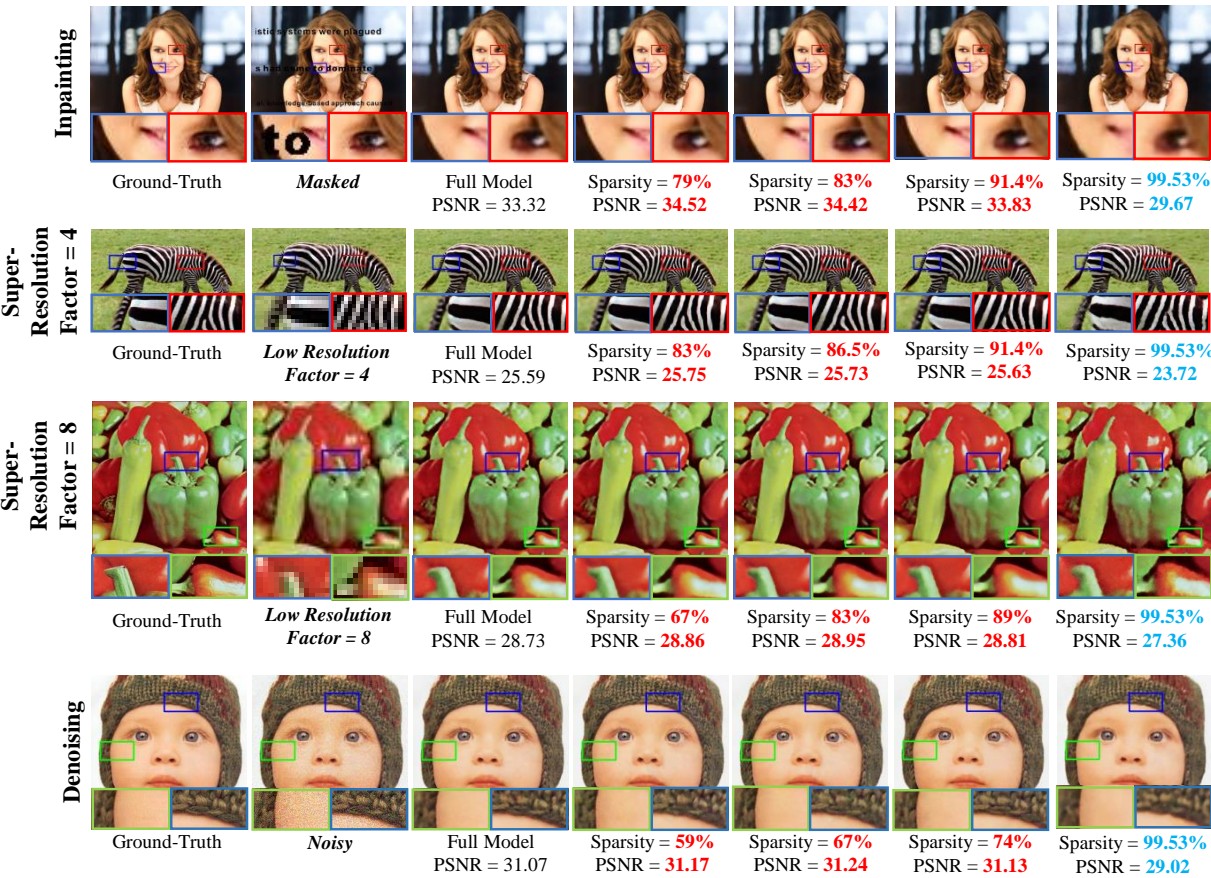

Figure 2: **LIP** visual results: inpainting (row 1), super-resolution (rows 2/3) and denoising (row 4). The last column (in blue) intends to display the results with the most extremely sparse subnetwork.

## 2.2 Under-parameterized Image Priors

Recall that, classical image regularizers in the spatial or frequency domains often rely on no or little learning component. For example, Tomasi & Manduchi (1998) first proposed a bilateral filtering method to smooth images while preserving edges by combining image values in a nonlinear way. After that, Sardy et al. (2001) proposed a wavelet-based estimator with a robust loss function to recover signals from noises. Later, Dabov et al. (2007b) proposed a strategy based on enhancing the sparsity through grouping similar 2-D image fragments into 3-D data arrays, followed by a collaborative filtering procedure. And this algorithm achieves state-of-the-art performances in terms of both peak signal-to-noise ratio and subjective visual quality. In addition to these traditional methods, Cao et al. (2008) proposed a Gaussian mixture model for image denoising that partitions an image into overlapping patches and estimates the noise-free image patch parameters using expectation maximization (EM) and Minimum mean square error (MMSE) techniques. Elad & Aharon (2006) also presented an approach to remove Gaussian noises based on sparse and redundant representations using trained dictionaries obtained with the K-SVD algorithm.

As another important concern, while over-parameterized DIPs are prone to overfitting the corrupted image, they can generate almost uncorrupted images surprisingly after a few iterations of gradient descent. Inspired by those, Heckel & Hand (2018) pioneered to design a concise and non-convolution neural network as the "prior for image restoration tasks. Such under-parameterized DIPs not only improve the efficiency of DIP-based restoration but also enable researchers to theoretically analyze the signal process. Besides, they find the architecture's simplicity could effectively avoid overfitting in the restoration process. Heckel & Soltanolkotabi (2019) further analyzed the dynamics of fitting a two-layer convolutional generator to a noisy signal and prove that early-stopped gradient descent denoises/regularizes.

## 2.3 Lottery Ticket Hypothesis

The lottery ticket hypothesis (LTH) Frankle & Carbin (2018) states that the dense, randomly initialized DNN contains a sparse matching subnetwork, which could reach the comparable or even better performance by independently being trained for the same epoch number as the full network do. Since then, the statement has been verified in a variety of fields. In natural language processing (NLP), Gale et al. (2019) established the first rigorous baselines for model compression by evaluating state-of-the-art methods on transformer Vaswani et al. (2017) and ResNet He et al. (2016). Later, Chen et al. (2020a) successfully found trainable, transferrable subnetworks in the pre-trained BERT model. In reinforcement learning (RL), Yu et al. (2019a) confirmed the hypothesis both in NLP and RL tasks, showing that "winning ticket" initializations generally outperform randomly initialized networks, even with extreme pruning rates. In lifelong learning, Chen et al. (2020b) first demonstrated the existence of winning tickets via bottom-up pruning. In graph neural networks (GNNs), Chen et al. (2021c) first defined the graph lottery ticket and developed an unified GNN sparsification (UGS) framework to prune graph adjacency matrices and model weights. And in adversarial robustness research, Cosentino et al. (2019) evaluated the LTH with adversarial training and they successfully proved this approach can find sparse, robust neural networks.

Simultaneously, many researchers started to rethink the network pruning techniques and build a rigorous baseline benchmark Liu et al. (2018); Evci et al. (2019). Some researchers explore the sparse subnetwork at the initialization stage (e.g., Wang et al. (2020) proposed the "GraSP" algorithm, which leverages the gradient flow.). And some try to find the winning tickets at early iterations Frankle et al. (2020d); Savarese et al. (2019). For example, You et al. (2019) found the "early-bird" tickets via low-cost training schemes with a large learning rate. However, as the original LTH paper Frankle & Carbin (2018) said, the proposed method cannot scale to large model and datasets. Rewinding was proposed by Frankle et al. (2019) to solve this dilemma. The found matching subnetworks also demonstrate transferability across datasets and tasks Morcos et al. (2019); Desai et al. (2019). Ma et al. (2021) thought the rewinding stage is not the only way to solve this problem and proposed a simpler yet more powerful approach called "Knowledge Distillation ticket".

## 3 Preliminaries and Approach

### 3.1 Finding Lottery Tickets

**Networks** For simplicity, we formulate the dense network output as $f(z, \theta)$, where $z$ is the input tensor and $\theta \in \mathbb{R}^d$ is the model parameters. In the same way, a subnetwork is defined as $f(z, m \odot \theta)$ with the binary mask $m \in \{0, 1\}^d$, where $\odot$ means element-wise product.

---

**Algorithm 1** Single Image-based IMP

> **Input:** The desired sparsity $s$, the random code $z$, the untrained model $f_u$.
> **Output:** A sparse DIP model $f(z; \theta \odot m)$ with image prior property.
> **Initialization:** Set $m_u = 1 \in \mathbb{R}^{||\theta||_0}$. Set iteration $i = 0$, training epochs $N$ and $j \in [0, N]$.
> **while** the sparsity of $m_u < s$ **do**
>     1. Train the $f_u(z; \theta_0 \odot m_u)$ for $N$ epochs;
>     2. Create the mask $m'_u$;
>     3. Update the mask $m_u = m'_u$;
>     4. Set the model parameters: $f(z; \theta_j)$;
>     5. create the sparse model: $f(z; \theta_j \odot m_u)$;
>     6. $i{+}{+}$;
> **end while**

---

**Algorithm 2** Weight-sharing IMP

> **Input:** The desired sparsity $s$, the random code $z$, the untrained model $f_u$, $\tilde{x}$ denotes the degraded image and images from $n$ domains $x_a \in \{x_1, x_2, ..., x_n\}$.
> **Output:** A sparse DIP model $f(z; \theta \odot m)$ with image prior property.
> **Initialization:** Set $m_u = 1 \in \mathbb{R}^{||\theta||_0}$. Set iteration $i = 0$, training epochs $N$ and $j \in [0, N]$.
> **while** the sparsity of $m_u < s$ **do**
>     1. loss $= \sum_{a=1}^n E(f(z; \theta \odot m); \tilde{x}_a)$;
>     2. Train the $f_u(z; \theta_0 \odot m_u)$ by Backpropagation (loss) for $N$ epochs;
>     3. Update the mask $m_u = m'_u$;
>     4. Set the model parameters $f(z; \theta_j)$;
>     5. create the sparse model $f(z; \theta_j \odot m_u)$;
>     6. $i{+}{+}$;
> **end while**

---

**Pruning Methods** We use the classic *iterative magnitude-based pruning* (IMP) method Frankle & Carbin (2018), which iteratively prunes the 20% of the model weight each time. In each IMP iteration, models are trained towards the standard DIP objective to fit the *degraded observations* for a certain number of training steps following the original DIP Ulyanov et al. (2018). Our basic algorithm performs IMP over just one degraded image, and the algorithm is summarized in Algorithm 1. We further design an extended algorithm, that can perform IMP for DIP over multiple degraded images, through backbone weight sharing: the algorithm is outlined in Algorithm 2 (neither algorithm requires clean images). To show the non-triviality of the identified *matching* subnetworks, we also compare LIP with random pruning and SNIP Lee et al. (2018), a pruning-at-initialization method. We also derive a new method based on empirical observations to decide when to stop IMP iterations to find matching networks with maximal sparsities without any reference to the clean ground truths. More details are in Appendix B.

**Experimental Setup** For evaluation models, we use hourglass architecture Ulyanov et al. (2018) and deep decoder Heckel & Hand (2018), as two representative untrained DNN image priors in the over- and under-parameterization ends, respectively. For evaluation datasets, we use the popular Set5 Bevilacqua et al. (2012) and Set14 Zeyde et al. (2010). We also evaluate the transferability of subnetworks on image classification datasets such as ImageNet-20 Deng et al. (2009) and CIFAR10 Krizhevsky et al. (2009). For metrics, we mainly compare the PSNR and/or SSIM results between the restored image and the ground truth, as in Fig. 11.

The parameter count of the original DIP model is 2.2 million (M); and that of the deep decoder is 0.1 M for denoising and super-resolution experiments, and 0.6 M for inpainting experiments, all following the original settings of Heckel & Hand (2018). The model sizes are plotted as horizontal coordinates in the figures. **We run all experiments with 10 different random seeds: every solid curve is plotted over the 10-time average, and the accompanying shadow regions indicate the 10-time variance. Most plots see consistent results across random seed experiments and hence small variances.** All images used are summarized in Fig. 17.

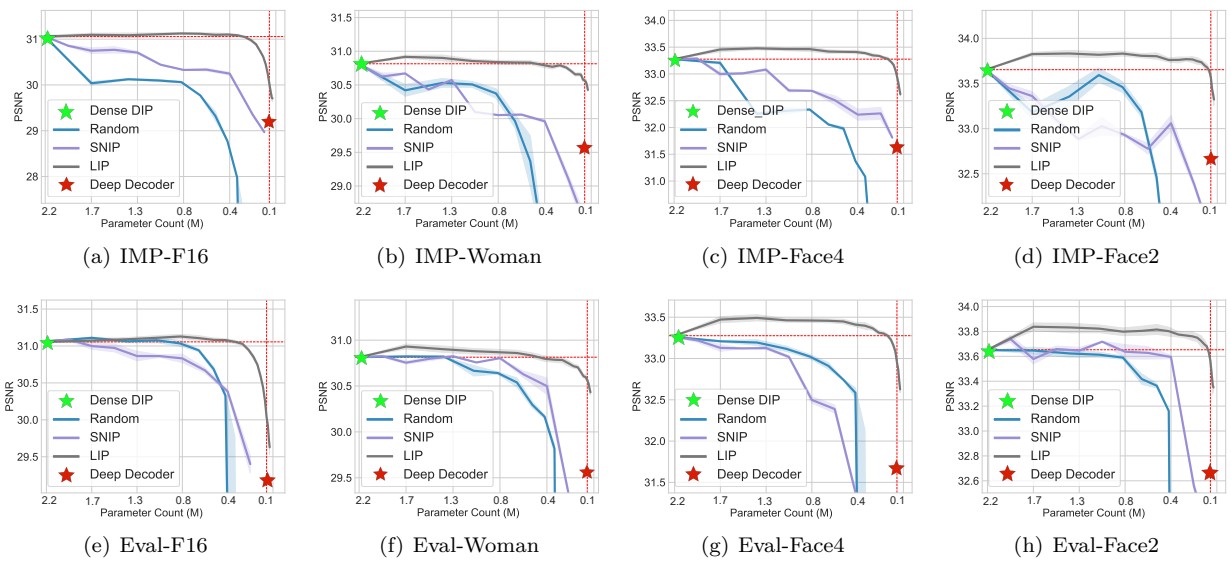

Figure 3: Experimental results of finding LIP subnetworks. The first row of the figure summarizes the LTH IMP training loops and the second row denotes the evaluation of found LIP. Note that we compare the LTH IMP with Random Prune (Random) and SNIP Lee et al. (2018) prune methods, on images from different (F16 and Woman) or same domains (Face2 and Face4). Background task is denoising.

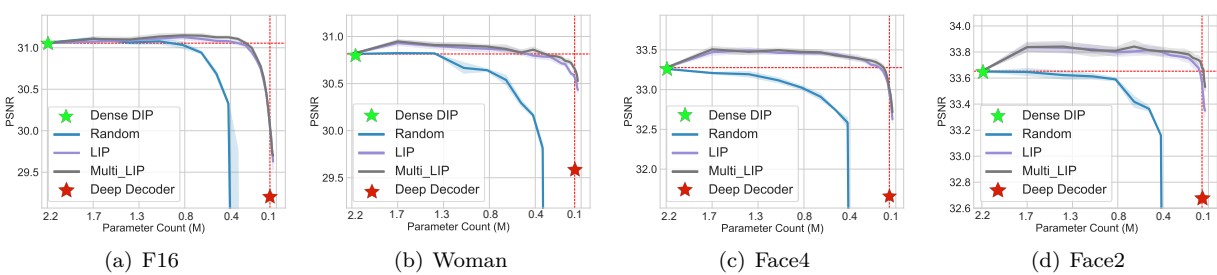

Figure 4: Experimental Results of Multi LIP on images from same/different domains. We compare Multi LIP with LIP and random prune methods. The background task is denoising.

## 4 LIP in Untrained Image Priors

**Existence of LIPs** In over-parameterized image priors, we first find the matching subnetworks with LIP property by implementing the single-image IMP on the DIP model. We apply the implemented Algorithm 1 on Set5 and Set14 to obtain the sparse subnetworks[2] and evaluate these subnetworks on the denoising task. Results of single-image IMP in Fig. 3 (curves in gray colors) verify the existence of LIPs. To be specific, during the IMP finding process, we can find the LIP subnetworks on untrained DIPs at sparsity as high as 86.58%. While at the evaluation stage, the found LIP subnetworks with the modified objectives are still applicable, matching the dense performance at sparsity as high as 83.23%. We also compare the single-image IMP with Random Pruning, and SNIP Lee et al. (2018). We observe from the first row in Fig. 3 that single-image IMP outperforms them at a wide sparsity range [20%, 96%].

**Is Multi-image IMP A Good Extension for Finding LIPs?** Like the original DIP, single-image IMP over-fits an untrained DNN on a single image and learns the features in that specific image during iterative magnitude pruning loops. To obtain the LIP subnets with more general features, we propose a new *multi-*

---

[2]We prune 20% of the remaining weights in each IMP iteration, resulting in sparsity ratios $s_i = 1 - 0.8^i$.

*image IMP* (Algorithm 2) for DIP where we replace the DIP objective with the average of multiple images. Note that all images will share the same fixed random code during IMP.

We evaluate the multi-image IMP in two different settings: (i) *cross-domain* setting where we apply the multi-image IMP to the five images from Set5 Bevilacqua et al. (2012); (2) *single-domain* setting where we apply the multi-image IMP to five images of human faces with glasses. We think images from Set 5 are more diversified because they include bird, butterfly and human face contents. We compare single-image IMP winning tickets found on the F16 and the Woman images from Set5 with the cross-domain ticket, and the single-image IMP winning tickets found on Face-4 and Face-2 images with the single-domain ticket. Results presented in Fig. 4 show that multi-image LIP subnetworks can achieve better performances than dense DIP models and randomly pruned ones in the cross-domain setting.

**To What Extent Can LIP tickets Be Transferred?**  In this part, we evaluate the transferability of LIP for DIP models from three perspectives, i.e., *across images*, *across image restoration task types*, and *from low-level to high-level tasks*.

*Observation 1: LIP can transfer across images.* As we obtained the multi-image LIPs from Set5, we evaluate them on different images in Fig.4. For instance, Fig. 4(a) shows the evaluation results on F16.png and we found the multi LIPs perform comparably well (better at extreme sparsities) with LIP dedicatedly found on the F16.png (the same phenomenon is reflected on the face-4.png (Fig.4(c)) and face-2.png (Fig.4(d))). This shows that LIP has reasonable transferability across images, even for those coming from slightly different domains.

*Observation 2: LIP can transfer across image restoration tasks but not to other high-level tasks.* We conduct experiments to verify if a LIP matching subnet identified on one restoration task can be *re-used* in another. Furthermore, if the above question has an affirmative answer, is such transferability sustainable when transferring to or from some high-level tasks such as classification?

To evaluate the transferability of LIP between image restoration tasks, we first find three LIPs for the denoising, inpainting and super-resolution tasks respectively and then transfer between them, as shown in Fig. 5. We observe that a LIP winning ticket transferred from another image restoration task always yields restoration performance comparable with the single-image LIP found on the original task, sometimes even better, for examples in Fig. 5(a) and 5(g).

We then evaluate the transferability of LIP between the denoising task and image classifications on CIFAR-10 and ImageNet-20 datasets. We show the results of transferring the denoising LIP to ImageNet-20 in Fig. 6(c) and CIFAR-10 in 6(a); the CIFAR-10 LIPs to denoising task in Fig. 6(b) and ImageNet LIPs to denoising in Fig.6(d). We find transferring denoising LIP to classification is unsuccessful. Moreover, transferring winning tickets on CIFAR-10 back to the denoising DIP task also fails to generate winning tickets that are comparable with denoising LIPs.

**Why is the LIP Subnetwork Special and Good?**  To answer this, we investigate the inner structures of different subnetworks in Fig. 7. The structure of the LIP subnetwork is drastically different from those found by SNIP and random pruning, in particular the distribution of layer-wise sparsity ratios. LIP tends to preserve weights of the earlier layers (closer to the input), while pruning the later layers more aggressively (e.g, Fig. 7(a)). In contrast, SNIP tends to prune much more of the earlier layers compared to the later ones. Random pruning by default prunes each layer at approximately the same ratio. Comparing the three methods seem to suggest that for finding effective and transferable LIP subnetworks, specifically keeping more weights at earlier layers is important. That is an explainable finding, since for image restoration tasks, the low-level features (color, texture, shape, etc.) presumably matter more and are more transferable, than the high-level features (object categories, etc.). The earlier layers are known to capture more low-level image features, hence contributing more to retraining the image restoration performance with DIP.

We summarize and plot the lawyer-wise sparsity ratio result of classification tickets in Figure 7(c) and Figure 7(f). Note that these 6 tickets are directly pruned by LTH on CIFAR10 dataset with the sparsity ratio of 36%, 59%, 89% and 95%. In contrast to the inner architecture of LIP subnetworks, we have observed an interesting trend in LTH that it tends to preserve more weights in the middle layers and prune weights in the

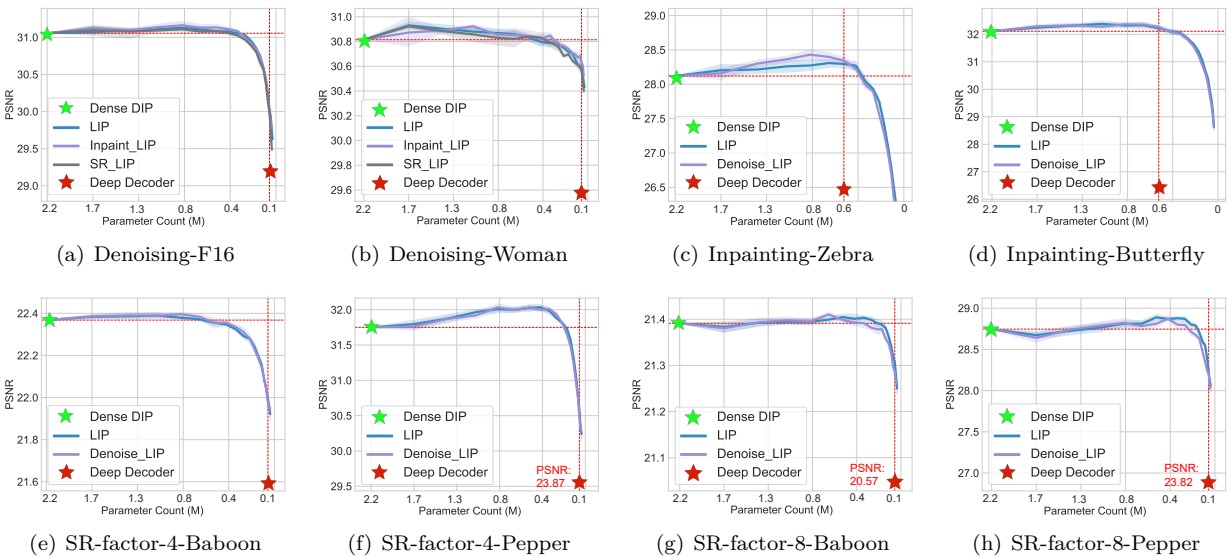

Figure 5: Transferability (cross tasks) experimental results. We study the transferability of denoising LIP on the restoration tasks such as inpainting and super-resolution (SR); we also study the inpainting and SR LIP on the denoising task. We consider two SR scale factors = 4, 8. We evaluate Multi-LIP subnetworks here.

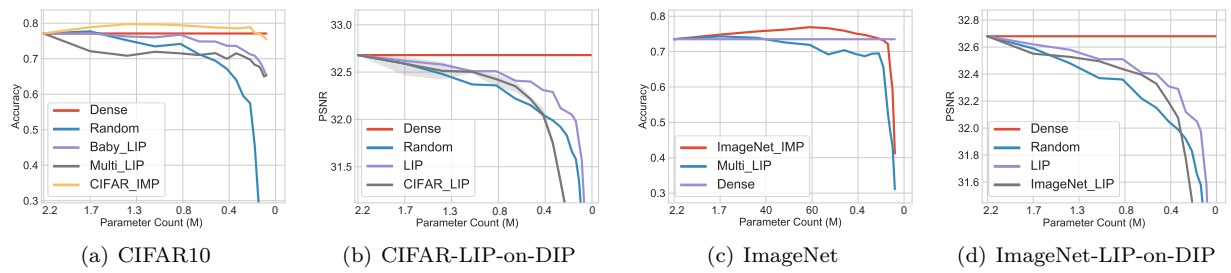

Figure 6: High- and low-level task transferability experiments. We test the denoising LIP on ImageNet-20 (a subset of ImageNet with images resized to 256*256) and CIFAR10 datasets. Note that we replace the last convolutional layer of DIP models with the linear layer and load the same initial weights. We also evaluate the classification LIP on the denoising task.

earlier and later layers (closer to input and output). This phenomenon may partially explain the challenges in transferring LIP subnetworks from restoration tasks to classifications, and vice versa.

The failure of transferring LIP subnetworks to image classification could also be partially explained by the aforementioned sparsity distribution discrepancy: LIP subnetwork tends to prune more weights from later layers, which might damage the network's capability in capturing semantic features (usually needing mid-to-high network layers). However, we observe that the bottleneck layers with presumably higher levels of pruning/sparsity will carry more contextual information and would lead to better downstream classification. Also, we could finetune on top of the intermediate layers of the model when more level information is needed. We assume that if we want the DIP model to transfer better to the image classification task, adding and finetuning linear layers on top of the intermediate layers (i.e., transferring only the winning tickets of the encoder) will help.

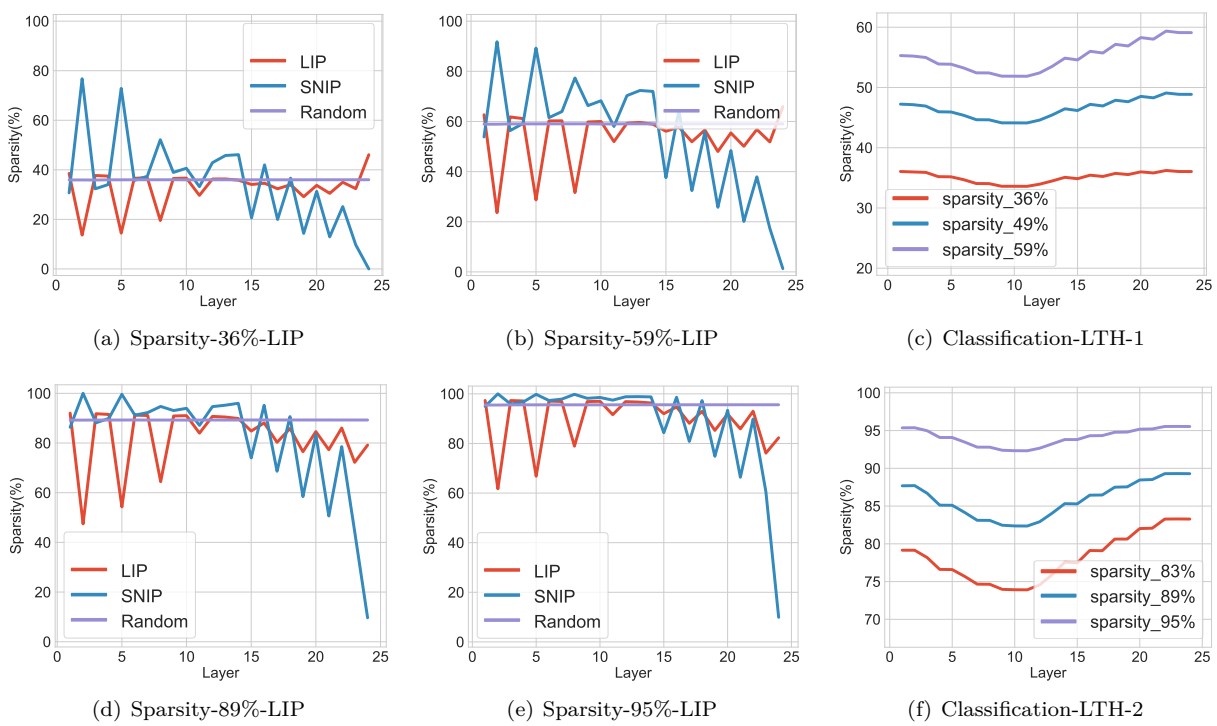

Figure 7: Layer-wise sparsity ratio results of LIP, SNIP, randomly pruned tickets, and classification tickets directly pruned by LTH. Note that we summarize the sparsity ratio of each layer: the ratio of the number of parameters whose values are equal to zero to the number of total parameters of the layer. And the x-axis of these figures is composed of the serial numbers of model layers. We sampled subnetworks with four different sparsities in LIP (sparsity $= 36\%, 59\%, 89\%, 95\%$), and six different sparsities in classification tickets (sparsity $= 36\%, 49\%, 59\%, 83\%, 89\%, 95\%$) to observe.

**Why do the LIP Subnetworks Outperform the Dense Model with fewer Parameters at the Beginning?**   Our initial conjecture is that pruning at low sparsities does not hurt the model capacity or only at a small level but rather enhances it with stronger robustness against noisy signals, a phenomenon that has been observed in Ulyanov et al. (2018).In fact, it has been widely observed that LTH-pruned subnetworks can outperform dense models in the low sparsity range, such as when 10% to 50% of parameters are pruned. This phenomenon was first observed in Frankle & Carbin (2018) and has been consistently observed in various machine learning tasks, spanning vision Chen et al. (2021b), languages Chen et al. (2020a), and reinforcement learning Yu et al. (2019a).

# 5   Extending LIP to Pre-trained Neural Network Priors

Although the implementation process of the GAN CS task and the DIP restoration task is different, we still find them related and could be unified for two reasons: 1) current research on LTH consists of two main streams, networks with weights with randomly initialized weights Frankle & Carbin (2018) and networks with pre-trained weights Chen et al. (2021b); 2) network structures with random weights or pre-trained weights can be priors, which is the commodity we view as the two could be unified. However, we do not intend to put the GAN CS part in a parallel position to the DIP part. We just want to demonstrate the effectiveness of our pruning method as an "extension" from training-from-scratch deep image prior to using pre-trained neural networks as image priors.

**Compressive Sensing using Generative Models**   Compressive Sensing (CS) reconstructs an unknown vector from under-sampled linear measurements of its entries Foucart & Rauhut (2013), by assuming the

unknown vector to admit certain structural priors. The most common structural prior is to assume that the vector is $k$-sparse in some known bases Candes et al. (2006); Donoho (2006), and more sophisticated statistical assumptions were also considered Baraniuk et al. (2010). However, those priors are inevitably oversimplified to depict the high-dimensional manifold of natural images. Bora et al. (2017) presented the first algorithm that used pre-trained generative models such as GANs, as the prior for compressed sensing. As a prior, the pre-trained generator encourages CS to produce vectors close to its output distribution, which approximates its training image distribution. Significant research has since followed to better understand the behaviors and theoretical limits of CS using generative priors, e.g., Hand & Voroninski (2018); Bora et al. (2018); Hand et al. (2018); Kamath et al. (2019); Liu & Scarlett (2020); Jalal et al. (2020).

Table 1: Results of GAN LIP. We evaluate the LIPs found in PGGAN on the compressed sensing (CS) and the inpainting (I) restoration tasks. The results are based on celebA-HQ dataset Lee et al. (2020). Note that we use the MSE (Mean Squared Error, per pixel) to evaluate the LIP effectiveness and we also compare the performance of LIP with random pruning results.

| Sparsity | 0% | 20% | 36% | 49% | 59% | 67% | 74% |
|---|---|---|---|---|---|---|---|
| **Random-CS** | 0.0725 | 0.0963 | 0.1165 | 0.1276 | 0.2184 | 0.2086 | 0.3655 |
| **LIP-CS** | 0.0725 | 0.0744 | 0.0732 | 0.0737 | 0.0711 | 0.0728 | 0.0728 |
| **Random-I** | 0.0541 | 0.0682 | 0.0748 | 0.08101 | 0.1142 | 0.1904 | 0.2195 |
| **LIP-I** | 0.0541 | 0.0542 | 0.0504 | 0.0514 | 0.0506 | 0.0524 | 0.0509 |

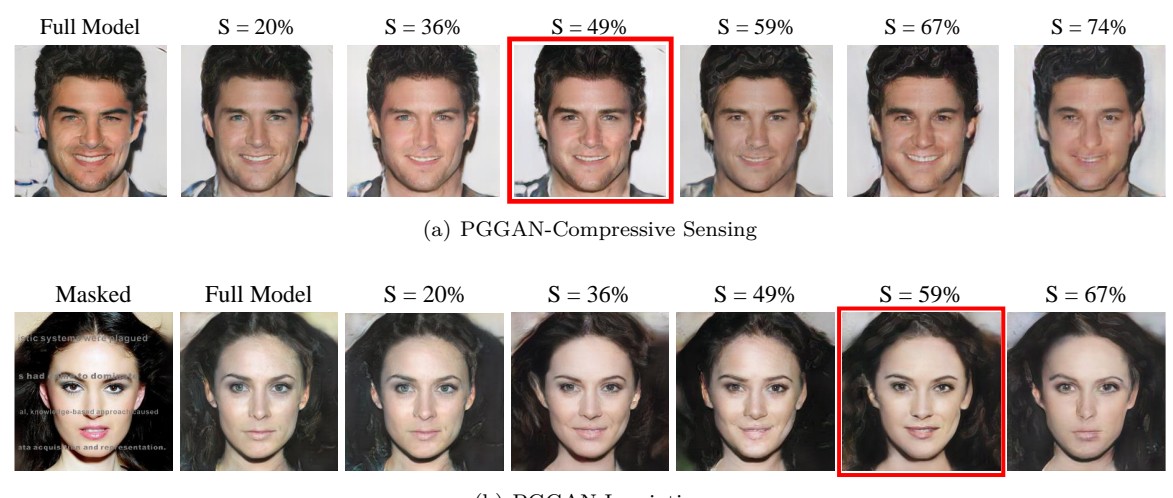

(a) PGGAN-Compressive Sensing

(b) PGGAN-Inpainting

Figure 8: Visual results of compressed sensing and inpainting using LIPs. The images framed in red are the reconstruction results of sparse subnetworks, which are better than those of the full model.

**Existence of LIP in GANs for Compressive Sensing** We use PGGAN Karras et al. (2017) pre-trained on CelebA-HQ dataset Lee et al. (2020) as the model in this section. To obtain LIP tickets in pre-trained GANs, we apply the IMP algorithm. In each IMP iteration, PGGAN is first fine-tuned on 40% of images in CelebA-HQ for 30 epochs, has 20% of its remaining weights pruned, and then reset to the pre-trained weights. We only prune the generator in the IMP process because it is found in Chen et al. (2021d) that pruning discriminator only has a marginal influence on the quality of the winning tickets. We then evaluate the tickets on the compressive sensing task following the setting in Jalal et al. (2020): we fix the number of measurements to 1,000 with 20 corrupted measurements, and minimize the MOM objective (Median-of-Means, an algorithm proposed by Jalal et al. (2020)) for 1,500 iterations to recover the images. We compare the performance (measured in per-pixel reconstruction error) of LIP with the dense baselines in the first

row of Table. 1 and provide a visual example in Fig. 8(a). Tickets with higher sparsities can match the reconstruction performance of the dense model, confirming the existence of the winning tickets.

**Transfer to other image restoration tasks – inpainting**   Besides the experiments on the compressed sensing restoration tasks, we also evaluate the effectiveness of GAN LIP on the inpainting task: masking the image and then optimizing the input tensor of the generator in the GAN LIP range to reconstruct the pristine image. More formally, consider the input tensor $z' \in \mathbb{R}^{1 \times 512}$, the pristine image $x$ sampled from CelebA-HQ, inpainting mask $A$ (binary mask), masked image $y = Ax$ and a generator $G$, then the optimization loss function is: $||AG(z) - y||_2$. The results are summarized in Table. 1 and Fig. 8(b), demonstrating the transferability of GAN LIP.

## 6   Limitations and Future Work

Although the LIP subnetworks outperform the dense DIP model and Deep Decoder on many image restoration tasks, we also find some limitations of LIP. Firstly, the LIP subnetworks cannot directly transfer from low-level (image restorations) to high-level vision tasks (image classifications) and we conjecturally attribute this to the sparsity distribution discrepancy in the inner structure of LIP subnetworks. Secondly, LIP subnetworks fail to achieve comparable performances of the dense DIP model when at extreme sparsities (as high as 99.53%). Thirdly, we observe that the LIP subnetworks do not perform very well on specific restoration tasks such as super resolution because of the inherent difficulty for the deep image prior models to generate high-quality textures. In the future, we will keep exploring more efficient ways to construct more sparse and trainable image prior models with powerful transferability.

## 7   Conclusion

This paper identifies the new intermediate solution between the under- and over-parameterized DIP regimes. Specifically, we validate the lottery image prior (LIP), a novel, non-trivial extension of the lottery ticket hypothesis to the new domain. We empirically demonstrate the superiority of LIP in image inverse problems: those LIP subnetworks often compare favorably with over-parameterized DIPs, and significantly outperform deep decoders under comparably compact model sizes. They also possess high transferability across different images as well as restoration task types.

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

## A   More Discussions of the Experiments

**Images Used in Experiments**   In Fig. 17, we organize and present the images used in this paper with their names. Note that these images are sampled from Set5 Bevilacqua et al. (2012) and Set14 Zeyde et al. (2010) datasets, and we use their default names. Note that these names are used in Section 4 (in curves and analyses). Also, in Table 5, we conduct the evaluation experiments on BM3D Dabov et al. (2007a), CBM3D Dabov et al. (2007a) and Set12 Zhang et al. (2017) datasets for fair comparisons with NAS-DIP and ISNAS-DIP models.

**Parameter Redundancy Problem**   The commonly used hourglass model in DIP Ulyanov et al. (2018) is highly complex in its parameters. The statistic results of parameter numbers is summarized in Table 3. We compare the parameter numbers of dense DIP models with the identified winning tickets using LIP and discover that the winning tickets could perform better than the full model while containing 2 million parameters fewer. This phenomenon motivates us to suspect that there is a high possibility of finding the matching subnetworks of the pristine dense DIP model, which indicates that the subnetworks may also contain the outstanding image prior property as the dense one does.

As to the under-parameterized image prior networks (e.g., deep decoder), we also count the number of non-zero parameters in Table 2 for a fair comparison in restoration tasks with LIP networks. As shown in the table, for denoising and super-resolution tasks, the number of parameters of the deep decoder is 0.1 million. For inpainting tasks, the parameter number is 0.6 million. Therefore, for denoise and super-resolution tasks, we choose the sparsity 97.19% of LIP for comparisons; for inpainting tasks, we choose the sparsity 73.70% for comparisons.

Table 2: Compare the number of non-zero parameters in Deep Decoder (DD)Heckel & Hand (2018) and our LIP tickets. Note that we compare these models on three restoration tasks: Denoising (DN), Super-resolution (SR) and Inpainting (IP). And 'LIP - 97.15%' means the sparsity of the LIP subnet is 97.15%.

| Model Structure | DD - 128 (DN, SR) | DD - 320 (IP) | LIP - 97.15% (DN, SR) | LIP - 97.19% (DN, SR) | LIP - 73.79% (IP) | LIP - 79.03% (IP) |
|---|---|---|---|---|---|---|
| Non-zero Parameter Numbers | 100224 (0.1M) | 619200 (0.6M) | 102746 (0.1M) | 90923 (0.09M) | 608298 (0.6M) | 494251 (0.5M) |

Table 3: Evaluation of the dense and LIP models on denoising task with the image bird. The LIP tickets achieve comparable results as the dense model when in extreme sparsity (parameter number: 2.2 million vs 0.2 million).

| | Dense Model | LIP Matching Subnetwork |
|---|---|---|
| Parameter Numbers (non-zero) | 2.2 Millon | **0.2 Millon** |
| Performance (PSNR) | 30.35 | **30.61** |

**Definition of Subnetworks and Finding Them**   Consider a network $f(x; \theta)$ parametered by $\theta$ with input $x$, then a subnetwork is defined as $f(x; m \odot \theta)$, where $\odot \in \{0, 1\}^d$, $d = ||\theta||_0$ and $\odot$ is the element-wise product. Let $\mathcal{A}_t^\mathcal{T}(f(x; \theta))$ to be the training algorithm, that is, training model $f(x; \theta)$ on the specific task $\mathcal{T}$ with $t$ iterations. We also denote the random initialization weight as $\theta_0$ and the pre-trained weight as $\theta_p$; $\theta_i$ as weight at the $i$-th training iteration and $\mathcal{E}^\mathcal{T}(f(x; \theta))$ the model performance evaluation. Following the definitions of Frankle et al. (2020a), we define that if the subnetworks is *matching*, it satisfies the following conditions (we use $\theta_p$ for example, to denote a pre-trained lottery ticket Chen et al. (2020a; 2021b); $\theta_0$ can be defined likewise):

$$\mathcal{E}^\mathcal{T}(\mathcal{A}_t^\mathcal{T}(f(x; \theta_p))) \leq \mathcal{E}^\mathcal{T}(\mathcal{A}_t^\mathcal{T}(f(x; m \odot \theta))). \tag{1}$$

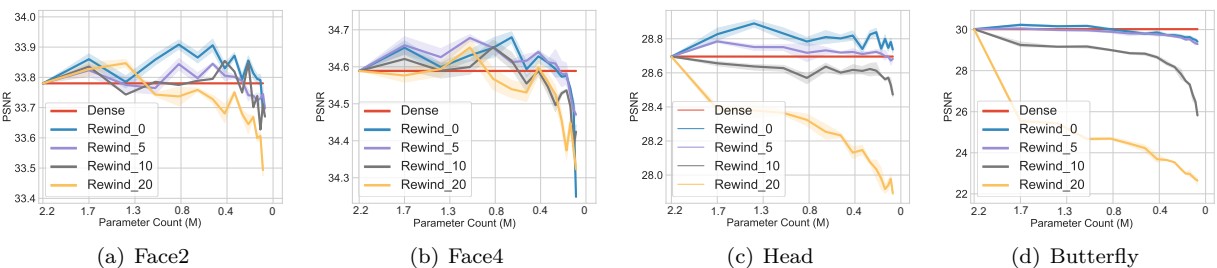

|   |   |   |   |
|---|---|---|---|
| (a) Face2 | (b) Face4 | (c) Head | (d) Butterfly |

Figure 9: Experiments of the rewind strategy (background task: denoising). Note that we train the model with $N$ epochs in IMP and Rewind_j means rewinding the ticket parameter to $\theta_j$, the weights after $j\% \times N$ steps of training.

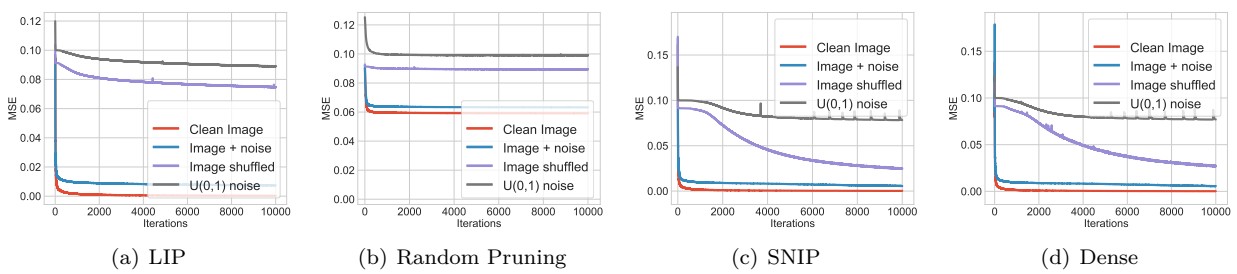

|   |   |   |   |
|---|---|---|---|
| (a) LIP | (b) Random Pruning | (c) SNIP | (d) Dense |

Figure 10: Learning curves using four different training targets: a clean image (Baby.png), the same image added with noises, the same randomly scrambled and white noise. Note that we use four different models: the LIP subnetwork ($S = 89\%$), randomly pruned subnetwork ($S = 89\%$), SNIP subnetwork ($S = 89\%$) and the dense model ($S = 0\%$). And we trained them in isolation in the same experimental settings for 10000 iterations.

That is a matching subnetwork that performs *no worse* than the dense model under the same training algorithm $\mathcal{A}^{\mathcal{T}}$ and the evaluation metric $\mathcal{E}^{\mathcal{T}}$. Similarly, we define the *winning tickets*: if a *matching* subnetwork $f(x; m \odot \theta)$ has $\theta = \theta_p$, then it is the *winning tickets* under the training algorithm $\mathcal{A}^{\mathcal{T}}$.

## B   Ablation Study of LIP Subnetworks

**The Effect of Weight Rewinding**   In this part, we study the effect of weight rewinding Frankle et al. (2019) when applied to the single-image IMP for DIP models. Weight rewinding is proposed to scale LTH up to large models and datasets. Specifically, we say we use $p\%$ weight rewinding if we reset the model weights at the end of each IMP iteration to the weights in the dense model after a $p\%$ ratio of training steps within a standard full training process, instead of the model's random initialization. For the single-image IMP in DIP, we consider 5%, 10% and 20% weight rewinding schemes. The resulting models are denoted as *Rewind_5*, *Rewind_10* and *Rewind_20*, respectively. The results of different weight rewinding schemes are summarized in Fig. 9. We can see that weight rewinding is not beneficial for identifying LIP in the DIP setting. Too much rewinding (10% and 20%) even hurts performance or fails it completely. We conjecture that this is due to the extremely low data complexity in DIP (single image).

**Learning Curves of Subnetworks with Different Training Targets**   To better explain the success of the LIP subnetworks, inspired by Figure 2 of Ulyanov et al. (2018), we further train the obtained subnetworks (LIP, SNIP and random pruning) with four different targets: 1) a natural image, 2) the same added with noise, 3) the same after randomly permuting the pixels and 4) the white noise. We also train the dense model as the baseline results. The experimental details are summarized in the caption of Fig. 10. We first observe that for LIP, SNIP, and dense models, the optimization converges much faster in case 1) and 2)

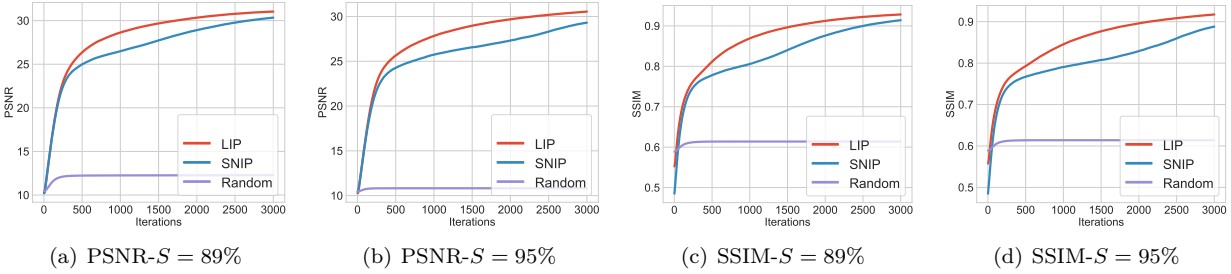

(a) PSNR-$S = 89\%$     (b) PSNR-$S = 95\%$     (c) SSIM-$S = 89\%$     (d) SSIM-$S = 95\%$

Figure 11: The learning curve plots when using different subnetworks towards the DIP task. In the figure, $S$ denotes the sparsity of the model. We compare both PSNR and SSIM values. For fair comparisons, we trained these subnetworks on the denoising task on the Baby image with 3000 iterations, then trained in isolation (the iteration number is recommended by Chen et al. (2020c) to capture the "early-stopping" phenomenon of DIP), and summarized their performances.

than in case 3) and 4). But the randomly pruned subnetworks have failed in all cases. Interestingly, we also find that SNIP subnetworks perform similarly to the dense model. Meanwhile, the parameterization of LIP subnetworks offers a higher impedance to noise and a lower impedance to signal than the dense model, which indicates that the separation of high frequency and low frequency is more obvious for winning network architectures.

**Does the Pruning Hurt the High-frequency Information of the Model?** Also, in order to observe whether the high-frequency information will be lost during the pruning, we apply Fourier Transformation to the Baby figure (described in Fig. 17) and visualize the frequency intensity of the ground-truth image and the reconstructions from three different subnetworks (LIP, SNIP and random pruning). The results are summarized in Fig. 18 and Fig.19. We found that compared with random pruning, LIP and SNIP can maintain most of the high frequency information in the ground-truth (e.g., in Fig 18, SNIP and LIP can both maintain the high-frequency information at the sparsity of 79%; however, SNIP could lose more high-frequency information than LIP at lower sparsity ratios.).

**Learning Curve Comparison of Using Various Subnetworks for the Restoration Task** We further compare the training convergence curves of different subnetworks on the restoration task. In Fig. 11, we summarize the convergence of LIP, SNIP and randomly pruned subnetworks on the denoising task, and the experimental details are included in the caption. We use the PSNR and SSIM metrics to measure the quality of the generated images: SSIM is often considered better "perceptually aligned", by attending more to the contrast in high-frequency regions than PSNR.

At the early stage of optimization, we observe that the learning curves of LIP and SNIP subnetworks are almost overlapped (either PSNR or SSIM curves), while the randomly pruned subnetworks failed to perform comparably with them. Yet when the iterations increase, the SNIP subnetworks start to lag behind the LIP subnetworks (e.g., the largest PSNR gap between the two can reach 3dB and the largest SSIM gap can be 0.7). Only the LIP subnetworks can match the comparable performances of the full model when reaching the 3000-th iteration. Lastly, the SSIM gap is noticeably enlarged at higher sparsity levels (95%) when comparing LIP and SNIP, which implies LIP is better at capturing perceptually aligned details.

**Visualization Results of the LIP Subnetworks** In Fig. 2, we present the restoration results of these subnetworks on tasks such as: denoising, inpainting and super-resolution (factor= $4, 8$). We successfully find the LIP subnetworks with high sparsities (e.g., 91.4% for inpainting, 89% for super-resolution and 74% for denoising), they achieve the comparable or better performance than the dense model. Moreover, we show the LIP subnetworks in the extreme sparsity (e.g., sparsity $s = 99.53\%$). We find that the performances of the LIP subnetworks are not largely degraded (e.g., the PSNR value decreases by 1.37 for super-resolution with factor 8.), which demonstrates the LIP subnetworks retain the image prior property of the dense models. In Fig. 12, we visualize the DIP learning process on the denoising task with the LIP subnetworks. And we also

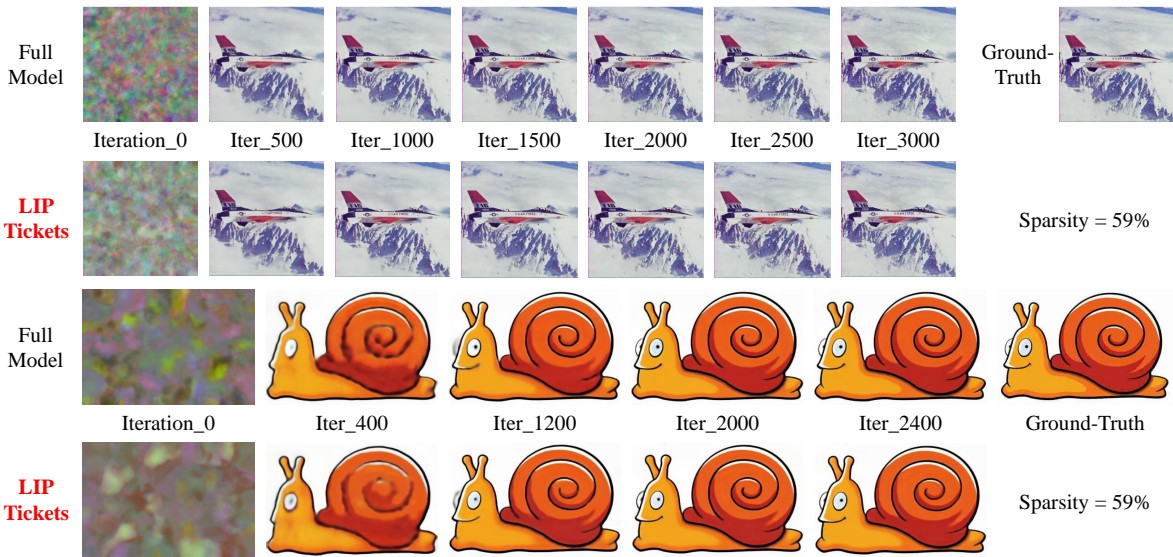

Figure 12: Visualize the learning process of the LIP subnetworks. We compare the results of the dense model and LIP tickets to study their learning ability toward one image. We use the default optimization iteration numbers (F16.png: 3000 iterations; Snail.png: 2400 iterations) in Ulyanov et al. (2018).

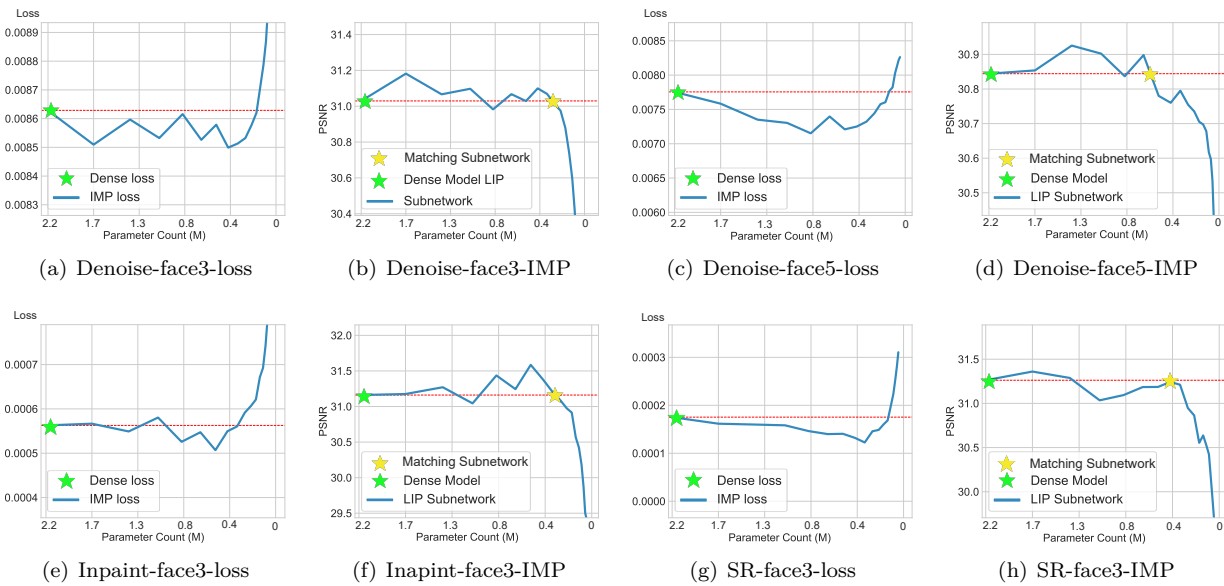

Figure 13: Study of early-stopping in IMP loops. For comparisons, we study the IMP loss values and the model performances during single-image IMP loops. We experiment with face3 and face5 images on three restoration tasks (denoising, inpainting and super-resolution). For comparisons, we plot the IMP loss and model performances during IMP loops. Note that we use the yellow star to denote the matching subnetworks and the green star for the dense model.

compare the learning ability of the dense model and LIP tickets. Interestingly, we find these subnetworks learn the general outline of the image at the early stage of the optimization process (e.g., the outline of the snail's shell at the first 500 iterations), and then they learn the detailed features of the image objects at the late stages (e.g., the eye features of the snail at the last 400 iterations). The outstanding learning ability also demonstrates their excellent image prior property.

**Comparisons with NAS-DIP Chen et al. (2020c) and ISNAS-DIP Arican et al. (2022) models.** Chen et al. (2022) first builds the searching space for upsampling and residual connections to capture better image priors for the dense DIP model. Arican et al. (2022) first finds that there is a small overlap of the best DIP architectures for different images and restoration tasks (e.g., denoising, inpainting and super-resolution). Therefore, they propose to do the NAS algorithm on DIP frameworks on specific images for computational savings. These methods aim to optimize the architecture of over-parameterized models to achieve better performances, and therefore, the model size of these two networks is still comparable to that of the dense DIP model (they neither compact the model nor prune its layers). In Table 5, we compare the performances of LIP subnetworks (the parameter number is 0.6 million) with DIP, NAS-DIP and ISNAS-DIP models. We find that LIPs can achieve comparable or even better performances than NAS-DIP models (especially on denoising and inpainting tasks). The ISNAS-DIP model gains state-of-the-art performances on restoration tasks such as denoising and inpainting. But when it comes to the super-resolution, LIP subnetworks and the NAS-DIP model outperform the ISNAS-DIP network.

**More discussions about LIP, NAS-DIP, and ISNAS-DIP.** From an algorithmic perspective, we would like to humbly emphasize that this work is the first attempt to investigate LTH in the DIP scenario, which could help to understand how the topology and connectivity of CNN architectures encode natural image priors, and whether "overparameterization + sparsity" make the best DIP recipe. This is a different angle from that of the NAS-based methods, which is about looking for the best "architecture (with dense weights)" for DIP.

In terms of implementation, it is worth mentioning that the published implementations of both NAS-DIP and ISNAS-DIP do not offer an out-of-the-box mechanism for regularizing the network size inherently during the searching process. In consideration of the limited duration of the response period, we refrained from extending the implementations of NAS-DIP and ISNAS-DIP independently. Our concern is that the ensuing search outcomes might diverge significantly from the concepts initially proposed in the original NAS-based works.

Methodologically, our pruning-based LIP is proposed to be orthogonal to NAS-DIP and ISNAS-DIP. More explicitly, the iterative pruning procedure can be equivalently applied to any network architecture discovered by the NAS-based methods, thus yielding a more compact network representation. We assume the feasibility of this combination to be grounded in the observed existence of the Lottery Ticket Hypothesis (LTH) across a range of network architectures including (a) compact Multilayer Perceptrons (MLP) and smaller convolutional networks Frankle & Carbin (2018); (b) over-parameterized networks with residual connections, such as ResNet and VGG networks Frankle et al. (2019); and (c) U-Net-like hourglass convolutional networks, as evidenced in this study, based on which both NAS-based methods search architectures.

**Discussions about the "early-stopping" in the single-image based IMP loops.** Similar to the DIP optimization, the single-image based IMP also needs "early-stopping" during the iterative pruning loops, or the final subnetwork will overfit the noises. In Figure 13, we summarize and plot the IMP loss and model performances during the iterative magnitude pruning. Specifically, we experiment with face3 and face5 images on three restoration tasks (e.g., denoising, inpainting and super-resolution.). We find that the matching subnetwork (yellow stars in the figure) appears where the value of IMP loss continuously increases two or three times. Based on this observation, we summarize the approach as follows: *To robustly find the matching subnetworks when on the single noisy image IMP loops, one can stop the iterative magnitude pruning when observing the continuous increases in IMP loss values.* However, there may be other ways (based on other signals) to identify the matching subnetworks in the single-image based IMP loops. For example, one can observe the gradient flow or the magnitude of gradients in the IMP loops to find the inner links between these signals and the appearance of matching subnetworks, which will be our future works. So far, the IMP loss value has been the most obvious signal in training loops.

**Transferability study of the single-image based LIP subnetworks** We evaluate several single-image based LIP subnetworks (Baby-LIP, Bird-LIP, Woman-LIP and Butterfly-LIP) on the Baby.png and compare their performances with MUlti-LIP subnetworks. The experimental results are summarized in Fig.14. We found that when the sparsity level is low (0%-40%), the single-LIP subnetworks perform a little worse than

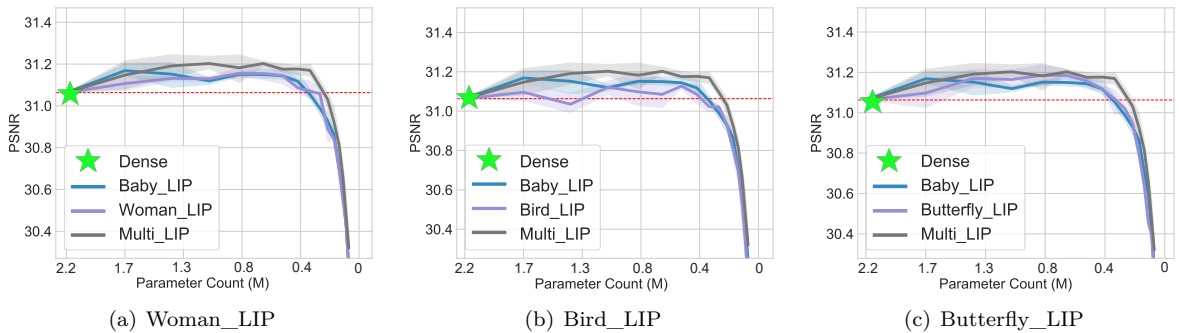

(a) Woman_LIP      (b) Bird_LIP      (c) Butterfly_LIP

Figure 14: Evaluate different LIPs on the baby.png. Note that the background task is denoising.

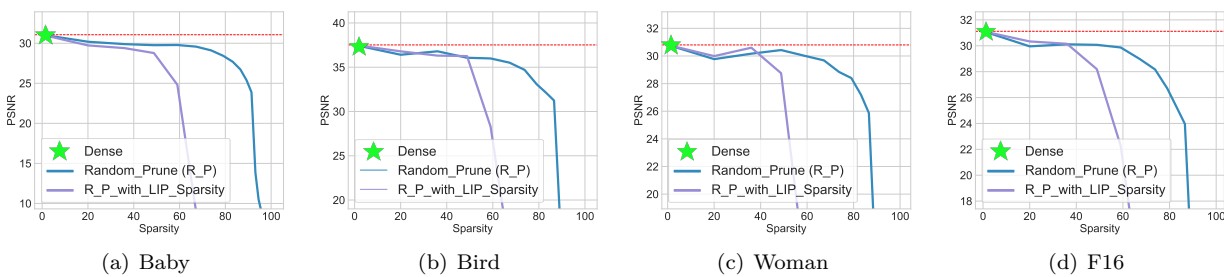

(a) Baby      (b) Bird      (c) Woman      (d) F16

Figure 15: Evaluate the random structure of the DIP models. We compare the performance of randomly pruned subnetworks with the ones randomly pruned with LIP sparsities. The background task is denoising.

the LIPs obtained on Baby.png. When the sparsity level increases (e.g., the number of remaining parameters is less than 0.6 million), they usually tend to perform comparably. Another intriguing phenomenon is that Multi-LIP performs comparably with the LIPs obtained on Baby.png but can perform better than the other LIPs when sparsity level is high, which also demonstrate that Multi-LIP could have better transferability than single-image based LIP subnetworks.

**Ablation Study on Random Structures of original DIP models** We have compared the randomly pruned subnetworks with LIP ones and have found these randomly pruned sparse networks perform worse, especially at high sparsity levels. This phenomenon indicates that the uniform randomness will lead to inferior performances. But what if we provide the LIP sparsity to the random structure? In Fig. 15, we summarize the experimental results on Baby.png, Bird.png, Woman.png and F16.png. We observed that the randomly pruned subnetworks with LIP sparsity will collapse at a much earlier stage than those that are uniformly randomly pruned. They may have comparable performances when at low sparsity levels (e.g. from 0% to 40%.). But the performance curve will quickly fall at high sparsity levels (e.g., from 50% to 90%.), which demonstrates that the inner structure of LIP subnetworks is unique.

**Comparison with Other State-of-the-art Pruning Methods** In the paper, we compare the effectiveness of LIPs with SNIP and random pruning. To better explore the effectiveness of LIP subnetworks, we further compare the sparse models with the "SynFlow" pruning methods Tanaka et al. (2020). The experimental results are summarized in Table.4. We observed that SynFlow subnetworks achieve comparable performances with the dense network and LIPs when at low sparsity levels (e.g., from 0% to 20%.). But LIP subnetworks start to win them out when the sparsity level increases (e.g., from 60% to 90%.). Also, the LIPs can achieve comparable performances when at extreme sparsities (e.g., when sparsity level equals to 80%.). But the "SynFlow" subnetworks have already failed. These phenomena indicate the effectiveness of proposed LIP methods, which also act as side support to Frankle et al. (2020b).

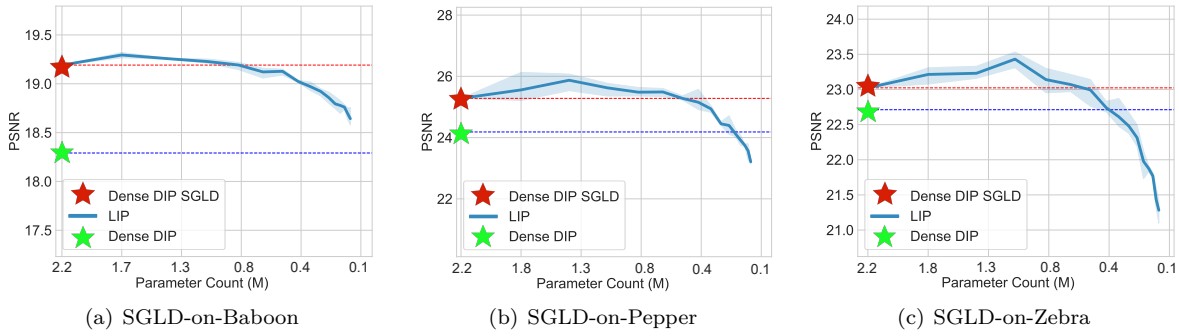

|  (a) SGLD-on-Baboon | (b) SGLD-on-Pepper | (c) SGLD-on-Zebra |

Figure 16: Experiment our method on the SGLD-DIP model Cheng et al. (2019). Note that we run this experiment with 3 different random seeds and the evaluation images are sampled from Set5 and Set14 datasets. Also, we mark the performance of the original dense DIP model in these figures (shown as the green star). The code of the SGLD-DIP model is available at `https://github.com/ZezhouCheng/GP-DIP`.

**How is the Performance of LIP Subnetworks on Other DIP-based Architectures?** To demonstrate the effectiveness and versatility of the LIP method, we have implemented the proposed pruning algorithm on the SGLD-DIP model Cheng et al. (2019), which is a state-of-the-art architecture based on the deep image prior. The SGLD-DIP model utilizes the stochastic gradient Langevin dynamics Welling & Teh (2011), enabling the model to mitigate the overfitting of noise and generate improved restoration outcomes. By applying the single-image IMP to this model, we conducted experiments with three different random seeds, using the images "Baboon.png", "Pepper.png", and "Zebra.png", all sourced from the Set5 and Set14 datasets. The experimental results, depicted in Figure 16, provide a summary of our findings. Particularly, we compare the performance of LIP SGLD models with the original dense models. Our observations reveal that the LIP SGLD models consistently outperform the dense networks across sparsity levels ranging from 20% to 79%, thereby illustrating the generality of our proposed methods.

Table 4: Comparison with synaptic-flow pruning method Tanaka et al. (2020). We compare the performances of LIPs and "SynFlow" subnetworks at various sparsity levels. Notice that the background task is denoising and the experiment image is F16.png.

| Sparsity (%): | 0 (dense) | 20 | 40 | 50 | 60 | 70 | 80 | 90 |
|---|---|---|---|---|---|---|---|---|
| **SynFlow** | 31.06 | 31.05 | 30.62 | 30.54 | 30.59 | 30.41 | 30.11 | 28.98 |
| **LIP** | 31.06 | 31.15 | 31.11 | 31.12 | 31.13 | 31.11 | 31.03 | 30.91 |

Table 5: Comparison results between LIP (sparsity = 73.79%, parameter number = 0.6M), DIP Ulyanov et al. (2018), NAS-DIP Chen et al. (2020c) and ISNAS-DIP Arican et al. (2022) models. The evaluation datasets are BM3D Dabov et al. (2007a), Set12 Zhang et al. (2017), CBM3D Dabov et al. (2007a), Set5 Bevilacqua et al. (2012) and Set14 Zeyde et al. (2010). And the results of DIP, NAS-DIP and ISNAS-DIP models are directly picked from Table 2 in Arican et al. (2022). Since NAS-DIP and ISNAS-DIP both aim to find better architectures to gain better model performances via NAS, their model sizes are comparable to that of the dense DIP network (there are no model compressions during the optimization). And we use "*" to denote this in the table.

| Datasets | DIP (2.2M) | LIP (0.6M) | NAS-DIP (*) | ISNAS-DIP (*) |
|---|---|---|---|---|
| Denoising | | | | |
| BM3D | 27.87 | 28.27 | 27.44 | **28.39** |
| Set12 | 27.92 | **28.12** | 26.88 | 28.06 |
| CBM3D | 28.93 | 29.45 | 29.13 | **30.36** |
| Inpainting | | | | |
| BM3D | 31.04 | 32.44 | 30.55 | **32.90** |
| Set12 | 31.00 | 31.78 | 30.86 | **32.22** |
| Super-resolution | | | | |
| Set5 ×4 | 29.89 | 30.52 | **30.66** | 30.22 |
| Set5 ×8 | 25.88 | **26.17** | 25.88 | 25.94 |
| Set14 ×4 | 27.00 | 27.31 | **27.36** | 27.19 |
| Set14 ×8 | 24.15 | **24.31** | 23.96 | 24.03 |

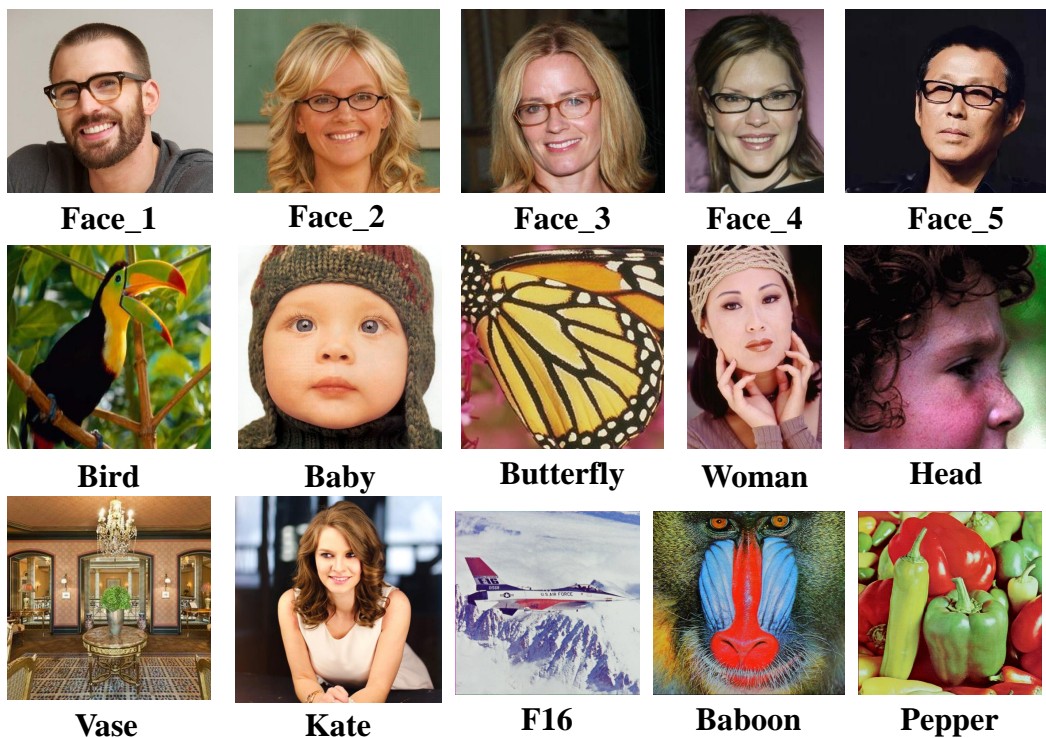

**Face_1**  **Face_2**  **Face_3**  **Face_4**  **Face_5**

**Bird**  **Baby**  **Butterfly**  **Woman**  **Head**

**Vase**  **Kate**  **F16**  **Baboon**  **Pepper**

Figure 17: Images used in plotting the curves of experiments.

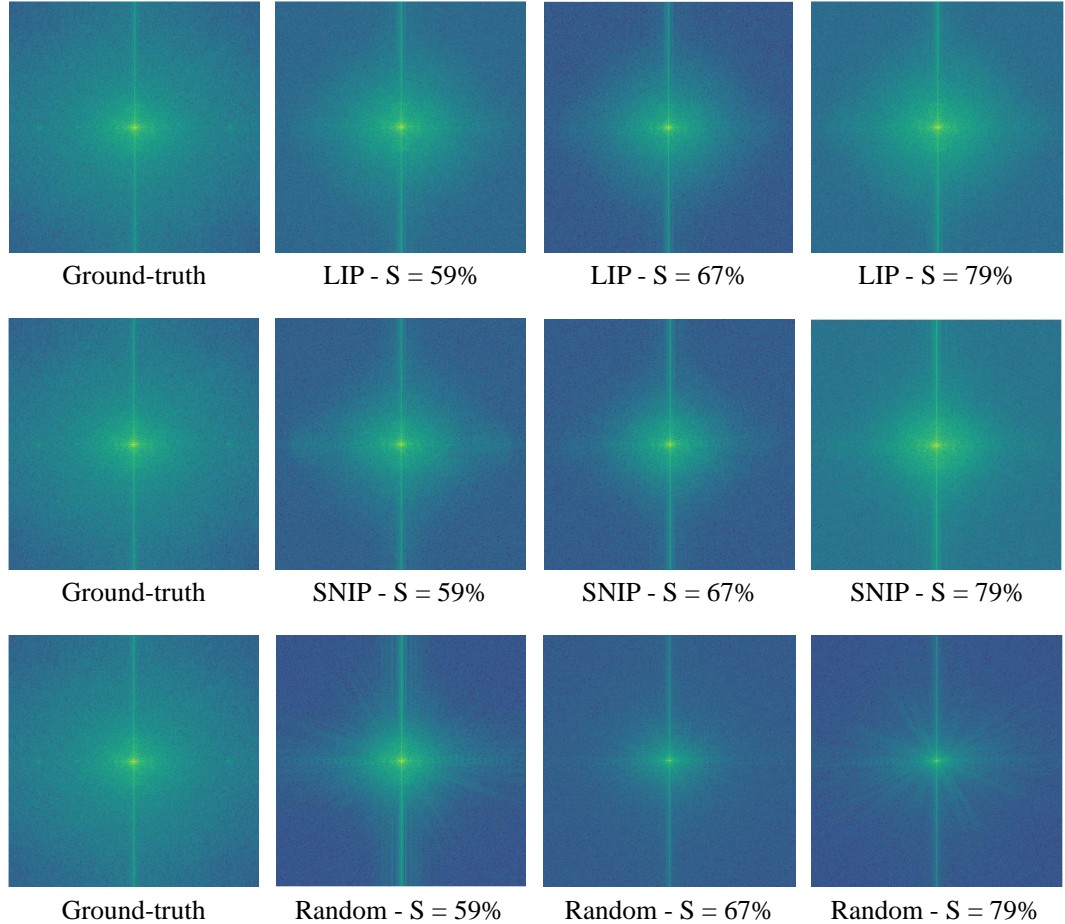

Figure 18: Evaluating the reconstruction images of different subnetworks (LIP, SNIP and random pruning) by FFT (Fast Fourier Transformation) to check whether the high-frequency information has been lost during pruning. Note that we experimented on the Baby.png. We found that compared with random pruning, LIP and SNIP can both maintain the high frequency information of the ground-truth. For example, the LIP and SNIP subnetworks both maintain most of the high-frequency information of the ground-truth at the sparsity 79%, but the LIP could also perform well at the sparsity 67% where the SNIP loses more high-frequency information.

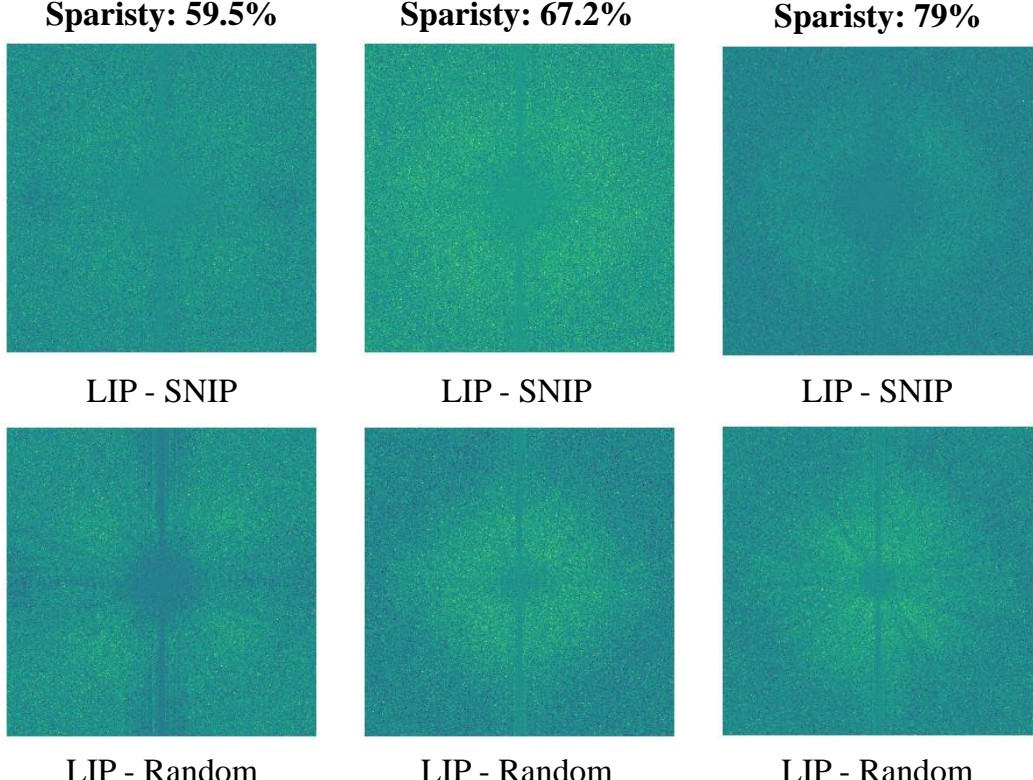

Figure 19: we visualize the difference maps between the FFT (Fast Fourier Transformation) results of reconstruction images generated by different methods (e.g., LIP, SNIP and random pruning). For example, 'LIP - random' means the difference map between the FFT results of restored images generated by LIP and randomly pruned subnetworks. The brighter color means a larger difference in values. Also, the center of the visualization images represents the image areas with frequency equal to zero and the surrounding parts denote areas with high frequency. We can observe that the reconstructions generated by LIP are consistent with those from other methods in the low-frequency domain (the center area of the FFT map) and mainly differ from other methods in the high-frequency domain.

