# OpenReview forum: "Chasing Better Deep Image Priors between Over- and Under-parameterization"
_TMLR — Accepted by TMLR_

### Review · Reviewer_RGN4 · 2023-04-09

**Summary Of Contributions:**

The paper proposed lottery image prior (LIP) that sits between over-parameterized deep image priors (DIP) and under-parameterized deep decoder. Analogous to the lottery ticket hypothesis (LTH), LIP aims to find a sparse sub-network from DIP that significantly out-performs under-parameterized image priors such as deep decoders while being compact in terms of model sizes. The key contribution of this paper is to verify the existence of LIP, which could be seen as a more compact DIP model (but not the same as deep decoders), not only out-performs deep encoders, but also free from severe over-fitting like DIP. By finding such LIP, better training efficiency could also be achieved.

**Audience:**

Yes

**Broader Impact Concerns:**

I don't find concerns on the ethical implications of the work that would require adding a Broader Impact Statement.

**Claims And Evidence:**

Yes

**Requested Changes:**

1. I think the background work section should be re-written, instead of listing a long list of papers that broadly belong to where this paper connects, the focus should focus more on how those work connect, influence or motivates the idea of LIP.
2. While I am impressed by the LIP performance showed in Figure 3, 4 and 5, as listed in the Strengths section, I notice that LIP out-performs DIP with less parameter counts at the beginning, and I hope to see more discussions from the authors on why this is the case, especially when we don't see the same performance in image classification tasks, as shown in Fig. 6.

**Strengths And Weaknesses:**

Strengths:
1. I think this paper is novel and high-quality.
2. The idea is interesting, and the writing is clear.
3. Experiment setup is nice - plotted on 10-time average with different random seeds and includes the corresponding variance.
4. The results are impressive (Fig.3 - 6).
5. Discussions on how LIP is different from SNIP (Fig. 7) is nice.

Weaknesses:
1. While I understand that the proposed LIP is closely connected to the over/under-parameterized image priors and LTH, and I appreciate the efforts that authors spent in listing a dense amount of references in the background work section, I don't find enough discussions on how those work connect to the proposed work. Especially in Section 2.3, where the authors simply list papers in LTH without any discussions on why those work belong to this paper.
2. Discussions on some of the experiment results are lacking (see requested changes section).

---

### Review · Reviewer_raKT · 2023-04-20

**Summary Of Contributions:**

In this article, the authors introduce a new "lottery image prior" (LIP), which leverages the innate sparsity of deep neural networks (DNNs) to pinpoint an intermediate, parameterized image prior that achieves superior performance, efficiency, and transferability. Specifically, the authors expand upon the lottery ticket hypothesis (LTH) to address low-level vision tasks such as image restoration and compressive sensing image reconstruction. Through a series of experiments, they not only demonstrate the presence of LIP in low-level vision tasks but also provide some interesting conclusions on the transferability of the proposed LIP.



**Audience:**

Yes

**Claims And Evidence:**

Yes

**Requested Changes:**

I recommend that the authors enhance the article by addressing the concerns raised in the identified weaknesses:

+ Provide more rigorous validation to demonstrate the reasons behind LIP subnetworks outperforming DIP full models in image restoration tasks.

+ The connection between GAN-based compressed sensing and Deep Image Prior (DIP) may seem ambiguous, leading to questions about the necessity of Section 5. Please clarify the significance of the GAN CS portion and justify the inclusion of Section 5 in the article.

+ Are there any failure cases or limitations associated with LIP? I would suggest the authors add a limitation analysis section in the revised article.

My current rating for the article is borderline. However, if the authors can address my concerns, I would be pleased to revise my rating upwards.

***Updated Rating***

I have read the authors' responses and comments from fellow reviewers. My major concerns can be successfully addressed. I am happy to recommend accepting this article.







**Strengths And Weaknesses:**

Pros:

+ First study on extending lottery ticket hypothesis (LTH) to address low-level vision tasks

+ Extensive empirical results are provided for several low-level image restoration tasks

+ Empirical results substantiate the existence of the proposed lottery image prior (LIP)



Cons:

- It is very interesting to observe that LIP subnetworks can surpass DIP models in various image restoration tasks. However, the underlying reasons remain unclear. The authors briefly mention that sparsity can enhance DNN robustness to noise and degradations, but the article lacks rigorous validation to support this claim.

- The relevance of GAN-based compressed sensing to Deep Image Prior (DIP) might appear unclear, raising the question of why Section 5 is necessary. While DIP and GAN-based compressed sensing follow distinct learning paradigms, employing the LIP for these two tasks may stem from different motivations. Thus, it is essential to explore and discuss these aspects in Section 5.

- This is a purely empirical study that extends LTH to address low-level vision tasks. From a technical standpoint, the contribution is marginal.

---

### Review · Reviewer_wwSi · 2023-05-07

**Summary Of Contributions:**

The paper proposes to prune the over-parameterized image priors to find a better performing and efficient image prior based on the Lottery Ticket Hypothesis. The pruned image prior results in better quality solutions to various inverse problems addressed by the DIP, with the relatively small number of parameters compared to the full model, of course outperforming efficient counterpart proposed in DeepDecoder. The paper presents various detailed studies including transferability across tasks, sensitivity study by the layer-wise sparsity ratio and application to pre-trained neural network prior using GAN.

**Audience:**

Yes

**Claims And Evidence:**

Yes

**Requested Changes:**

- Clarify the based DIP method used for applying the LTH and show the effectiveness of the method in the state of the art DIP method or architecture.
  - If possible, compare the effectiveness of the LTH in various DIP methods
- Reduce the network size by the competing methods such as NAS-DIP or ISNAS-DIP and compare the quality by them and the vanilla IMP that the LIP is using



**Strengths And Weaknesses:**

**Strengths**
- **S1**: First showing the LTH is applicable in DIP domain. The subnetwork performs better than the Deep decoder and even the original full size network when pruned.
- **S2**: Various detailed studies on transferability across tasks, sensitivity study by the layer-wise sparsity ratio and application to pre-trained neural network prior using GAN.

**Weaknesses**

- **W1**: The proposed idea is basically finding a better suited network architecture embedded in a large DIP network. The lottery ticket happens to be small in size but not guaranteed. The idea of finding better performing network architecture for DIP has been explored in NAS-DIP and ISNAS-DIP. Although these two methods did not try to reduce the size of the resulting networks, it is interesting to observe the performance when they try to reduce the size of the network.
- **W2**: Finding optimal size is an open question.
- **W3**: Lack of comparison to the state of the art DIP models. There are number of DIP models (Liu et al. 2019, Jo et al. 2021, Mastan & Raman 2020, 2021, Gandelsman et al. 2019, Yang et al. 2021) but there is no comparison to those models or at least try to use the best DIP model to prune the network. It is not clear why DIP method the method is mostly applied for the evaluation.

---

### Decision · Action_Editors · 2023-06-25

**Recommendation:** Accept with minor revision

**Comment:**

Please refer to the comments above.

**Audience:**

The researchers working in machine learning and computer vision will be interested in the findings of this paper.

**Claims And Evidence:**

This paper conducted the first study on extending the lottery ticket hypothesis (LTH) to learn deep image priors to address low-level vision tasks, and empirical results substantiate the existence of the proposed lottery image prior (LIP). The pruned image prior results in better quality solutions to various inverse problems addressed by the DIP, with the relatively small number of parameters compared to the full model. The idea is interesting in general, and well validated by the experiments. The paper is well written. Two of three reviewers vote for the acceptance, while one reviewer concerned with the comparison to the relevant competitors. For this concern, authors made some efforts in the rebuttal: the authors add more explanations about their algorithms V.S. LIP/NAS-DIP/ISNAS-DIP. The authours also implemented the pruning algorithm to some state-of-the-art models. Additionally, the authors explained that the proposed pruning-based LIP is orthogonal to NAS-DIP and ISNAS-DIP.

Overall, this paper has a good idea, which is well supported by extensive experiments. The AE suggested the acceptance of this paper, subjecting to some minor revision. In particular, for the camera ready version, it is suggested that some key discussion and conclusions about the proposed algorithm V.S. state-of-the-art models/algorithms, should be included in the main paper.